

# Guanidine thiocyanate solution facilitates sample collection for plant rhizosphere microbiome analysis

Xiaoxiao Sun[1], Meiling Wang[1], Lin Guo[1,2], Changlong Shu[1], Jie Zhang[1] and Lili Geng[1]

[1] State Key Laboratory for Biology of Plant Diseases and Insect Pests, Institute of Plant Protection, Chinese Academy of Agricultural Sciences, Beijing, China
[2] Department of Agronomy, Jinlin Agriculture University, Changchun, Jilin, China

Corresponding author
Lili Geng, llgeng@ippcaas.cn

## ABSTRACT

The interactions between rhizosphere microorganisms and plants are important for the health and development of crops. Analysis of plant rhizosphere bacterial compositions, particularly of those with resistance to biotic/abiotic stresses, may improve their applications in sustainable agriculture. Large-scale rhizosphere samplings in the field are usually required; however, such samples, cannot be immediately frozen. We found that the storage of samples at room temperature for 2 days leads to a considerable reduction in the operational taxonomic unit (OTU) number and the indices of bacterial alpha-diversity of rhizosphere communities. In this study, in order to overcome these problems, we established a method using guanidine thiocyanate (GTC) solution for the preservation of rhizosphere samples after their collection. This method allowed the maintenance of the samples for at least 1 day at room temperature prior to their cryopreservation and was shown to be compatible with conventional DNA isolation protocols. Illumina sequencing of V3 and V4 hypervariable regions of the 16S rRNA gene was used to assess the feasibility and reliability of this method, and no significant differences were observed in the number of OTUs and in the Chao and Shannon indices between samples stored at −70 °C and those stored in GTC solution. Moreover, the representation of *Pseudomonas* spp. in samples stored in GTC solution was not significantly different from that in samples stored at −70 °C, as determined by real-time quantitative polymerase chain reaction ($p > 0.05$). Both types of samples were shown to cluster together according to principal coordinate analysis. Furthermore, GTC solution did not affect the bacterial taxon profiles at different storage periods compared with those observed when storing the samples below −70 °C. Even incubation of thawed samples (frozen at −70 °C) for 15 min at room temperature induced minor changes in the bacterial composition. Taken together, our results demonstrated that GTC solution may provide a reliable alternative for the preservation of rhizosphere samples in the field.

## INTRODUCTION

Plant-associated bacteria, particularly rhizosphere bacteria, play important roles in plant growth. Some rhizosphere bacteria produce different compounds, such as nutrients and hormones, that promote the growth of plants; moreover, some are soil-borne pathogens that can lead to a dramatic yield loss, whereas others are neutral (*Blom et al., 2011*). Plant rhizospheres have been shown to contain approximately $10^{11}$ bacterial cells per gram of root, more cells than in the phyllosphere and soil (*Egamberdieva et al., 2008*). Plant-growth-promoting rhizobacteria (PGPR) are a large bacterial group, first identified as the bacteria enhancing plant productivity and immunity (*Lugtenberg & Kamilova, 2009*). However, several recent studies have demonstrated that PGPR induce physical and chemical changes in plants, increasing their tolerance to salinity (*Tank & Saraf, 2010*), drought (*Vurukonda et al., 2016*), cold (*Kakar et al., 2016*), and heavy metals (*Ullah et al., 2015*). Furthermore, PGPR may increase nitrate and phosphate uptake from soil (*Pii et al., 2015*), thereby helping reduce the use of fertilizer and water contamination. Most importantly, the investigation of the interactions between rhizosphere bacteria and plants will help promote their use in agriculture.

Currently, both culture-dependent and -independent methods are used for analysis of rhizosphere bacteria. In the past, denaturing gradient gel electrophoresis has been widely used as a culture-independent method for analysis of microbial community structure; however, high-throughput sequencing has been developed as a more sensitive technique and is now commonly used. Thousands of operational taxonomic units (OTUs) are obtained by high-throughput sequencing (*Lundberg et al., 2012*; *Bulgarelli et al., 2013*; *Turner et al., 2013*), whereas DGGE usually only yields dozens of OTUs (*Vallaeys et al., 1997*).

From the collection of samples to the sequencing of 16S rRNA genes, many factors, such as template preparation (*Tian et al., 2017*), storage conditions (*Flores et al., 2015*), amplification primers (*Klindworth et al., 2013*), and sequencing technologies (*Tremblay et al., 2015*), affect the alpha and beta diversity of bacterial communities of samples, and cause inaccurate analysis of rare taxa. However, the initial issue affecting microbiome analysis is variations in bacterial composition due to outdoor temperature. Previous studies have reported that delayed freezing has a considerable effect on bacterial communities in soil samples (*Rubin et al., 2013*) and human stool samples (*Flores et al., 2015*). Some reports have indicated that short-term incubation at room temperature has no obvious effects on the composition and diversity of microbial communities, particularly for human fecal samples (*Tedjo et al., 2015*; *Guo et al., 2016*). Several groups have reported that incubation at room temperature does not affect the overall bacterial community structure of soil samples (*Dolfing et al., 2004*; *Lauber et al., 2010*). However, to the best of our knowledge, few studies have evaluated these features in plant rhizosphere samples, which are more diverse than soil samples (*Lundberg et al., 2012*). Accordingly, it is reasonable to hypothesize that outdoor temperature may have different effects on plant rhizosphere samples. Immediate freezing of samples is usually not possible in the field, particularly in

remote areas. Additionally, the use of liquid nitrogen or ice in a portable insulated container is also not practical when large-scale samples are collected. Thus, novel methods are needed to overcome these problems.

Two recent reports indicated guanidine thiocyanate (GTC) solution promotes a stability in composition and diversity of human fecal microbial analysis at room temperature form 1 day to 1 week (*Nishimoto et al., 2016*; *Hosomi et al., 2017*). Guanidine isothiocyanate denatures RNase and DNase, inhibits the growth of bacteria (*Chomczynski & Sacchi, 2006*; *Hisada, Endoh & Kuriki, 2015*), and is widely used in DNA isolation (*Stearns et al., 2015*; *Neubeck et al., 2017*) and the preservation of human stool samples. However, this compound has not been applied for analysis of the plant rhizosphere microbiome to date.

In this study, we examined the effects of GTC solution on prevention of alterations in bacterial taxa and assessed the feasibility of this approach for analysis of the rhizosphere microbiome. The objective of this study was to determine whether GTC buffer could be used to preserve soil samples at room temperature for at least 1 day prior to additional cryopreservation. We expect that this method may have applications in sampling large-scale rhizosphere soil in fields without requiring immediate freezing. Here, we employed the Illumina MiSeq of 16S rRNA amplicon to assess the feasibility of using GTC solution, and attempted to develop a simple and reliable method for storing plant rhizosphere samples in field.

## MATERIALS AND METHODS

### Study system

Peanut (*Arachis hypogaea* L.), the world's fourth largest oilseed crop, is an allotetraploid (AABB, $2n = 40$) originating from *A. duranensis* (A genome) and *A. ipaensis* (B genome) (*Seijo et al., 2007*). Deciphering the rhizosphere microbiome of *A. hypogaea* L. can not only help us fully understand the plant–microbe interactions to improve plant health and yield, but may also provide insights into the control of subterranean insect pests (*Geng et al., 2018*).

### Rhizosphere compartment sampling

The peanut cultivar *Huayu22* was provided by Shandong Peanut Research Institute and cultivated in a greenhouse with a nutritional soil and vermiculite mixture (volume ratio 2:1, pH = 5.23, organic matter content = 222 g kg$^{-1}$, total nitrogen = 7.00 g kg$^{-1}$) as in our previous study (*Geng et al., 2018*), at 28 °C with 60% relative humidity and a 14-h photoperiod. Rhizosphere soil was sampled as previously described by *Bulgarelli et al. (2013)* with minor changes. Loose soil was removed from the roots by shaking. Peanut roots were cut off using sterile scissors, transferred to sterile 50-mL plastic centrifuge tubes containing 40 mL phosphate-buffered saline (PBS; pH 7.0, eight μL Silwet L-77), and soaked for 5 min. The samples were then vortexed for 15 s. Subsequently, roots were removed, and the rhizosphere soil suspension was passed through a sieve with 100 openings per square inch and centrifuged at 3,200×g for 15 min. Individual rhizosphere soil samples were collected from nine peanut plants.

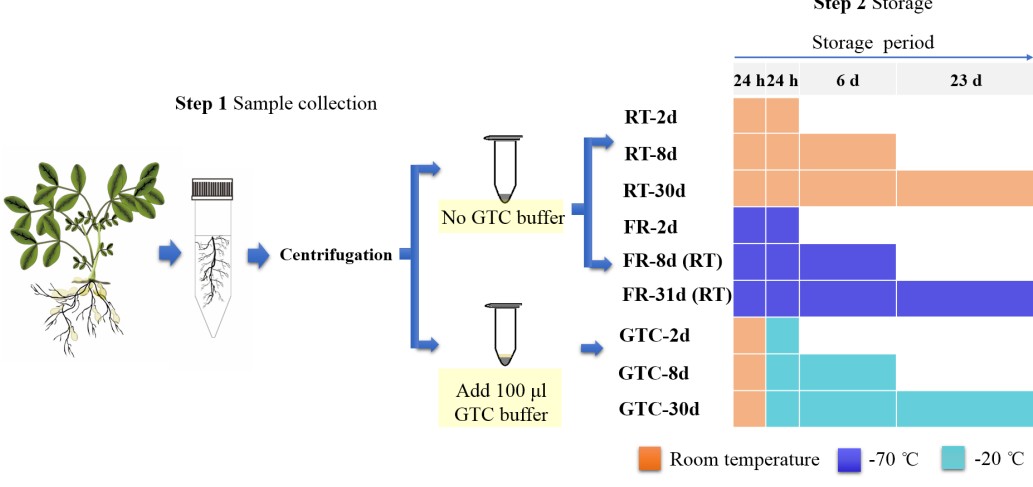

**Figure 1 Flow chart showing experimental design.** The samples were stored and processed under different conditions: RT samples, at room temperature; FR samples, at −70 °C; GTC samples, stored with 100 μL guanidine thiocyanate for 24 h at room temperature and maintained at −20 °C long-term. All samples thawed for 10 min and after that, FR-8d (RT) and FR-31d (RT) were kept in room temperature for another 15 min. DNA was isolated at day 2, 8, and 31.

Three rhizosphere soil samples from three plants were homogenized as one sample. Each sample was then divided into nine parts (100 mg each) and stored under different conditions (Fig. 1): (1) at room temperature (25 °C, RT sample); (2) frozen at −70 °C (FR sample); and (3) supplemented with 100 μL GTC buffer, containing 100 mM Tris–HCl (pH 9), 40 mM ethylenediaminetetraacetic acid, 4M GTC, and 0.001% bromothymol, stored at room temperature for 24 h, and then frozen at −20 °C (GTC sample). For the GTC and RT treatments, bacterial genomic DNA was extracted at different time points (days 2, 8, and 31 following the collection). For the FR samples, bacterial genomic DNA was extracted on day 2 after initiation of the experiment, and the thawed samples were kept at room temperature for 15 min before isolation on days 8 and 31.

## Extraction of bacterial genomic DNA

Genomic DNA was extracted from rhizosphere soils using a PowerSoil DNA Isolation Kit (MoBio Laboratories, Carlsbad, CA, USA) according to the manufacturer's instructions. For the GTC treatment, the suspension was centrifuged at 3, 200×$g$ for 15 min, and the GTC buffer was removed. Soil samples were washed 1 with one mL PBS and centrifuged at 5,000×$g$ for 15 min to remove PBS.

## GTC buffer and DNA extraction

The effects of GTC buffer on DNA extraction were analyzed. Rhizosphere soil samples were collected as described above and divided into 15 parts (100 mg each). The soil samples were treated as follows, with three repetitions per treatment: one of the treatments was stored at −70 °C without GTC buffer (N-GTC), and the other four treatments were stored at −20 °C after adding 100 μL GTC buffer. Before DNA extraction, GTC buffer

was removed by centrifugation at 3,200×*g* for 15 min and washed without PBS (GTC-W0) or washed once (GTC-W1), twice (GTC-W2), or three times (GTC-W3) with one mL PBS. Genomic DNA was extracted from the soil samples using a PowerSoil DNA Isolation Kit (MoBio Laboratories, Carlsbad, CA, USA) according to the manufacturer's instructions. The DNA concentration was determined using a NanoDrop 2000/2000c (Thermo Fisher Scientific, Rockford, IL, USA). Copy numbers of the 16S rRNA genes in these samples were determined by real-time quantitative polymerase chain reaction (qPCR). Data were analyzed using one-way ANOVA (Tukey's HSD) with SPSS version 19.0 (SPSS, Inc., Chicago, IL, USA).

## 16S rRNA sequencing and downstream data analysis

Next-generation sequencing library preparations and Illumina MiSeq sequencing were conducted at Biomarker, Inc. (Beijing, China). A part of the hypervariable region of 16S rRNA (V3 + V4 region) was amplified using universal primers with barcodes (Table S1, 341F: 5′-CCTAYGGGRBGCASCAG-3′, and 806R: 5′-GGACTACNNGGGTATCTAAT-3′). PCR was performed using Phusion High-Fidelity PCR Master Mix (New England Biolabs, Ipswich, MA, USA), and PCR products were purified using a Qiagen Gel Extraction Kit (Qiagen, Hilden, Germany). Sequencing libraries were prepared using a TruSeq DNA PCR-Free Sample Preparation Kit (Illumina, San Diego, CA, USA) following the manufacturer's protocol. The library quality was evaluated on a Qubit @ 2.0 Fluorometer (Thermo Fisher Scientific, Rockford, IL, USA) and Agilent Bioanalyzer 2100 system (Agilent Technologies, Palo Alto, CA, USA). The library was sequenced on an Illumina HiSeq 2500 platform (Illumina, San Diego, CA, USA), yielding 250-bp paired-end reads.

Quality-filtering using Trimmomatic (version 0.36) (*Bolger & Giorgi, 2014*) was performed with the following criteria: (i) bases off the start and end of a read below a threshold quality (score <3) were removed; (ii) and the reads were truncated at any site receiving an average quality score of less than 5 over a four-bp sliding window, discarding the truncated reads that were shorter than 100 bp. Clean reads were compared with Gold database (Broad Microbiome Utilities, version microbiomeutil-r20110519) using UCHIME, and chimeric reads were filtered to obtain the effective reads (*Edgar et al., 2011*). Sequences with ≥97% similarity were clustered into the same OTUs using USEARCH software (version 9.2.64) (*Edgar, 2013*). Representative OTU sequences were selected, and a 16S rRNA OTU table and abundance data were generated (Table S2). Additionally, these representative sequences were annotated to species according to the Ribosomal Database Project classifier (version 2.2) and the Greengenes (version 13.8) (*DeSantis et al., 2006*).

Alpha-diversity indices were calculated using USEARCH software and data were analyzed using one-way ANOVA (LSD) with SPSS version 19.0 (SPSS, Inc., Chicago, IL, USA) to compare various groups with each other. Weighted UniFrac distances were generated using relative abundance (*Lozupone & Knight, 2005*). Principal coordinate analysis (PCoA) and ANOSIM based on weighted UniFrac distances were performed

using R software (version 3.4.0). Pearson correlation coefficients were additionally calculated using the OTU abundance data and SPSS Statistics 24 (IBM, New York City, NY, USA) (*Nishimoto et al., 2016*). Fold changes in the taxonomic abundances of the top 15 genera between GTC/RT and FR-2d treatments were calculated, and the data were analyzed using Wilcoxon signed-rank test (SPSS Statistics 24; IBM, New York City, NY, USA).

## Quantitative real-time PCR

Real-time qPCR analysis was performed using a 7500 Real-Time PCR System (Applied Biosystems, Foster City, CA, USA) with SYBR Premix Dimer Eraser (Perfect Real Time; Takara Bio, Otsu, Japan). The reaction conditions were as follows: 94 °C for 15 min, followed by 40 cycles of 94 °C for 10 s and 61 °C for 32 s. Reactions were performed in triplicate. A ~140-bp fragment of the 16S rRNA gene was enzymatically amplified with primers 357F (5′-CTCCTACGGGAGGCAGCAG-3′) and 519R (5′-GWATTACCGCGGCKGCTG-3′) (*Lane, 1991*). Another fragment of the 16S rRNA gene of *Pseudomonas* spp. was amplified with a specific primer pair: Ps280F (5′-TAACT GGTCTGAGAGGATGATCAGT-3′) and Ps441R (5′-CCCAACTTAAAGTGCTTT ACAATCC-3′) (*Geng et al., 2018*). The relative abundance was calculated by dividing the number of copies of the 16S rRNA gene of *Pseudomonas* spp. by the total number of 16S rRNA gene copies for all species. Data were analyzed using one-way ANOVA (Tukey's HSD) with SPSS version 19.0 (SPSS, Inc., Chicago, IL, USA).

## RESULTS

### Effects of GTC buffer on DNA concentration

To explore the effects of GTC buffer on the DNA concentrations of rhizosphere samples, we designed five different treatments for implementation before total DNA isolation. Compared with the control, GTC buffer caused a 75.3% decrease in the DNA concentration ($p < 0.05$, Fig. 2A); washing two or three times with PBS also significantly reduced the DNA concentration ($p < 0.05$, Fig. 2A), whereas washing once with PBS did not affect the DNA concentration ($p > 0.05$, Fig. 2A). Copy number of the 16S rRNA gene transcript were determined by real-time qPCR. The results showed that washing once with PBS did not affect copy number of the 16S rRNA gene transcript, whereas the use of GTC buffer or washing two or three times with PBS led to a significant decrease in the copy number of the 16S rRNA gene transcript (Fig. 2B). Based on the above results, we washed the samples preserved with GTC with PBS once before extracting total DNA using the PowerSoil DNA Isolation Kit.

### Room-temperature storage affected the diversity of rhizosphere bacterial communities

Following quality control and assembly, 56,214–62,335 sequences of the V3 and V4 variable regions of the 16S rRNA gene were obtained from each sample, and data were deposited in the NCBI SRA database under accession number SRP134256 (Table 1).

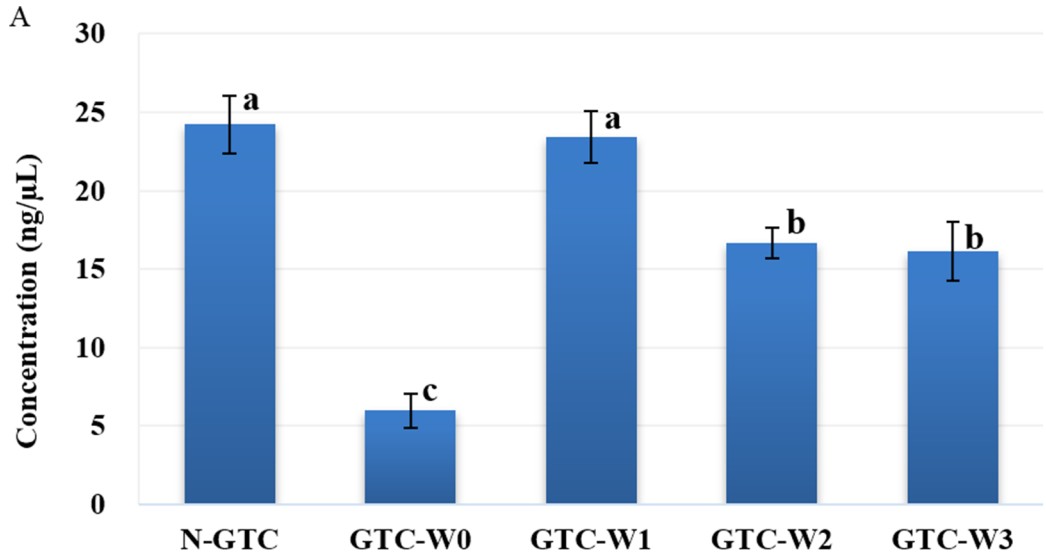

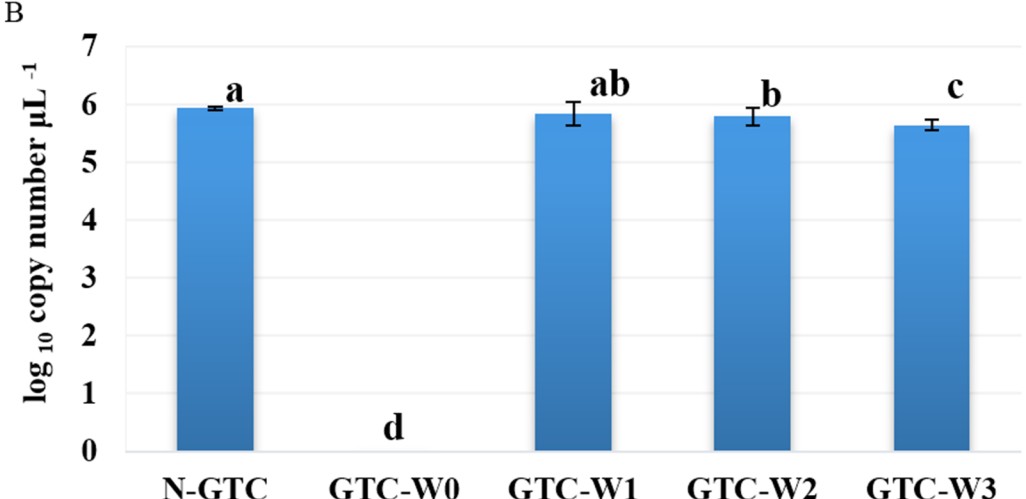

**Figure 2 DNA concentration (A) and number of copies of 16S rRNA gene (B) of peanut rhizosphere samples from five different treatments.** The five different treatments were performed as follows before DNA extraction: rhizosphere samples were collected and preserved in GTC buffer (named GTC-W0) and then washed with PBS buffer one (GTC-W1), two (GTC-W2), or three (GTC-W3) times; rhizosphere samples without GTC buffer were used as positive controls (N-GTC). $n = 3$. The error bars represent standard deviation. Different lower-case letters indicate significant differences between groups analyzed by one-way ANOVA (Tukey's HSD) ($p < 0.05$). Full-size 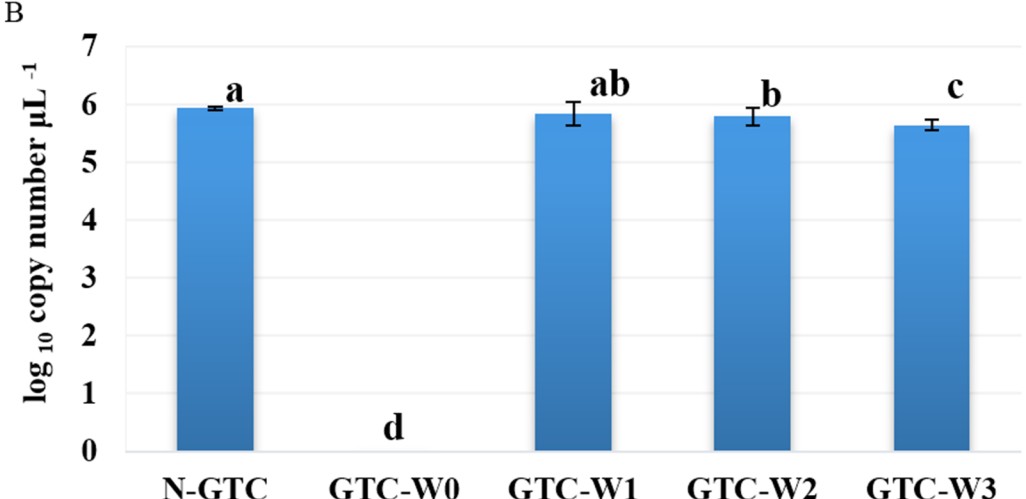 DOI: 10.7717/peerj.6440/fig-2

The relative abundances of the different taxa in all samples are presented in Table S2. The number of high-quality sequences for samples stored at room temperature ranged from 58,342 to 62,335. No significant differences in the number of high-quality sequences were observed between the RT-2d and FR-2d samples, but the average number of OTUs obtained for the RT group was found to be significantly lower than that obtained for the FR treatment (from 1,329 ± 128 to 2,154 ± 79; $p < 0.05$; Table 1). Compared with that obtained for the FR group, considerable decreases in the Chao and Shannon indices was observed in the RT group.

**Table 1 Pyrosequencing of 16S rRNA and alpha-diversity indices.**

| Treatment[*] | High-quality sequences | OTU number | Chao index | Shannon index | SRA database accession no. |
|---|---|---|---|---|---|
| **FR-2d** | $56,214.0 \pm 6,924.0^b$ | $2,066.0 \pm 155.0^b$ | $2,068.2 \pm 154.7^b$ | $8.3 \pm 0.1^b$ | SRP134256 |
| **RT-2d** | $58,342.0 \pm 399.0^{ab}$ | $1,347.0 \pm 63.0^c$ | $1,350.1 \pm 63.5^c$ | $7.0 \pm 0.1^d$ | |
| **GTC-2d** | $60,940.0 \pm 863.0^{ab}$ | $2,314.0 \pm 37.0^a$ | $2,316.0 \pm 37.1^a$ | $8.5 \pm 0.0^a$ | |
| **FR-8d (RT)** | $58,259.0 \pm 4,254.0^{ab}$ | $2,219.0 \pm 44.0^{ab}$ | $2,221.0 \pm 44.0^{ab}$ | $8.5 \pm 0.1^{ab}$ | |
| **RT-8d** | $62,145.0 \pm 646.0^a$ | $1,446.0 \pm 63.0^c$ | $1,450.3 \pm 62.7^c$ | $7.2 \pm 0.2^c$ | |
| **GTC-8d** | $60,640.0 \pm 273.0^{ab}$ | $2,236.0 \pm 48.0^{ab}$ | $2,237.4 \pm 48.4^{ab}$ | $8.4 \pm 0.1^{ab}$ | |
| **FR-31d (RT)** | $60,987.0 \pm 1,351.0^{ab}$ | $2,176.0 \pm 18.0^b$ | $2,178.0 \pm 17.9^b$ | $8.5 \pm 0.1^{ab}$ | |
| **RT-31d** | $62,335.0 \pm 322.0^a$ | $1,192.0 \pm 68.0^d$ | $1,195.9 \pm 67.6^d$ | $6.9 \pm 0.1^d$ | |
| **GTC-31d** | $61,742.0 \pm 1,489.0^a$ | $2,302.0 \pm 24.0^a$ | $2,303.6 \pm 24.4^a$ | $8.5 \pm 0.0^a$ | |

**Note:**

[*] Three independent biological replicates were analyzed in each group. Different lower-case letters indicate significant differences between groups analyzed by one-way ANOVA (LSD) analysis ($p < 0.05$).

## GTC and FR samples yielded similar taxonomic and abundance profiles

Guanidine thiocyanate was used at the final collection step and then removed by centrifugation before DNA isolation. An increase in the number of high-quality sequences between GTC-31d and FR-2d samples was observed. GTC-2d and -31d samples also showed a higher number of OTUs and the Chao and Shannon indices than that of FR-2d samples (Table 1; $p < 0.05$). Proteobacteria represented the dominant component of the peanut rhizosphere in all treatments (Table S2). Furthermore, fold changes in the taxonomic abundances of the top 15 genera between GTC and FR-2d treatments were calculated (Fig. 3C), and no significant differences were obtained in the Wilcoxon signed-rank test ($p = 0.226$, median = 0.086). However, significant changes were obtained at the genus level between RT and FR-2d groups (Fig. 3B; Wilcoxon signed-rank test, $p < 0.001$, median = −1.585). The relative abundance of *Pseudomonas* spp. in all samples was analyzed by real-time qPCR. The representation of *Pseudomonas* spp. in all GTC samples was not significantly different from that in FR-2d samples ($p > 0.05$, Fig. 4) but was significantly lower than that in all RT-2d and 31d samples ($p < 0.05$, Fig. 4). These results were consistent with the Illumina sequencing results.

Principal coordinate analysis was used to characterize differences in the rhizosphere bacterial community structure between all analyzed groups (Fig. 5), showing a clear clustering of the samples. FR-2d and all GTC samples were shown to cluster together, indicating similar bacterial community compositions, which was consistent with the previously presented Pearson's correlation analysis. The first principal coordinate (PC1) contributed to 60% of the total variation and revealed that the structure of the bacterial community in the RT samples differed from those in the FR-2d and GTC samples. The main factors in PC2 were found to be different time points in the RT samples, which contributed to approximately 19% of the variation.

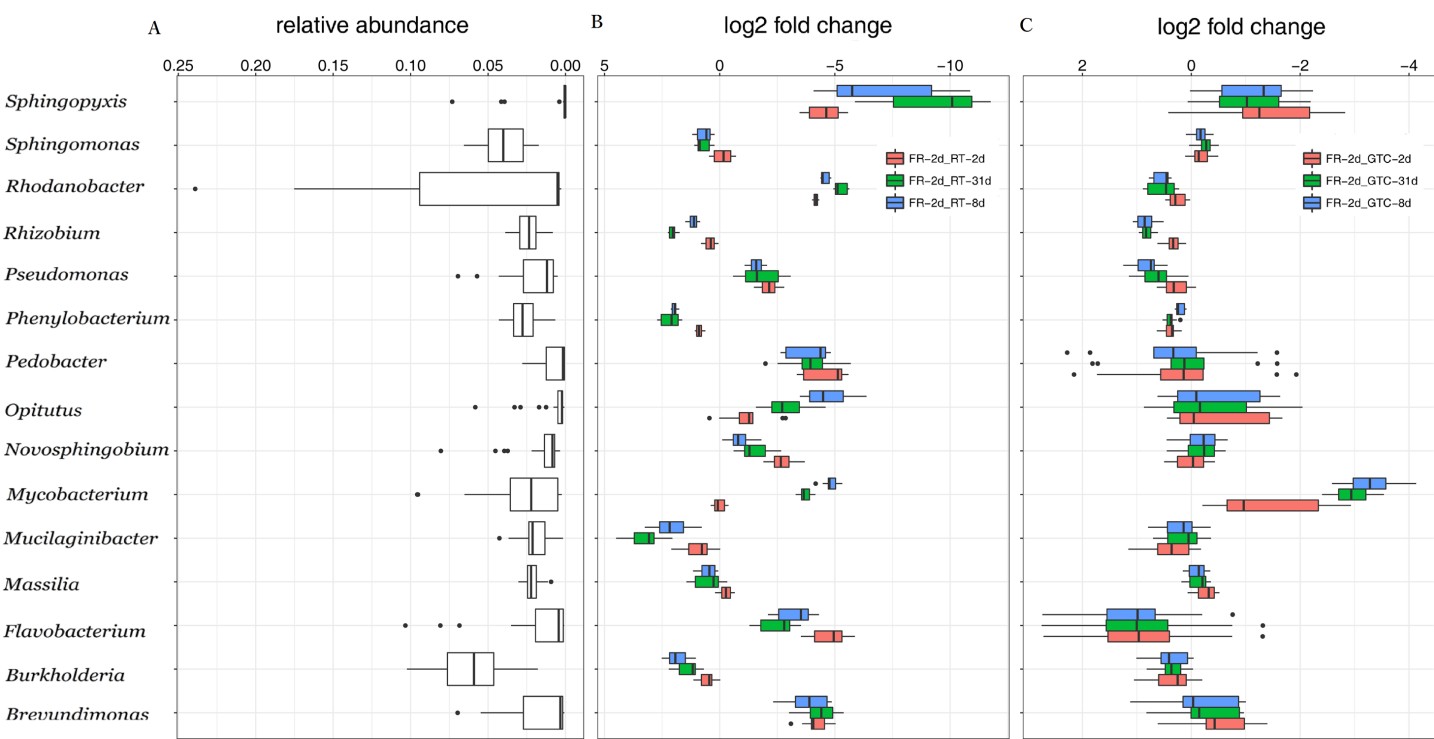

**Figure 3 The distribution of top 15 genera in the FR, GTC, and RT groups (A).** Comparisons of the taxonomic abundances of top 15 genera between FR and RT groups (B), and FR and GTC groups (C). FR-2d, rhizosphere samples stored at −70 °C for 2 days ($n = 3$). GTC-2d, -8d, and 31d are samples stored in 100 μL of GTC guanidine thiocyanate at room temperature for 24 h, and −20 °C for 1, 7, and 30 days, respectively ($n = 3$). RT-2d, -8d, and 31d are samples stored at room temperature for 2, 8, and 31 days, respectively ($n = 3$). The error bars represent standard deviation.

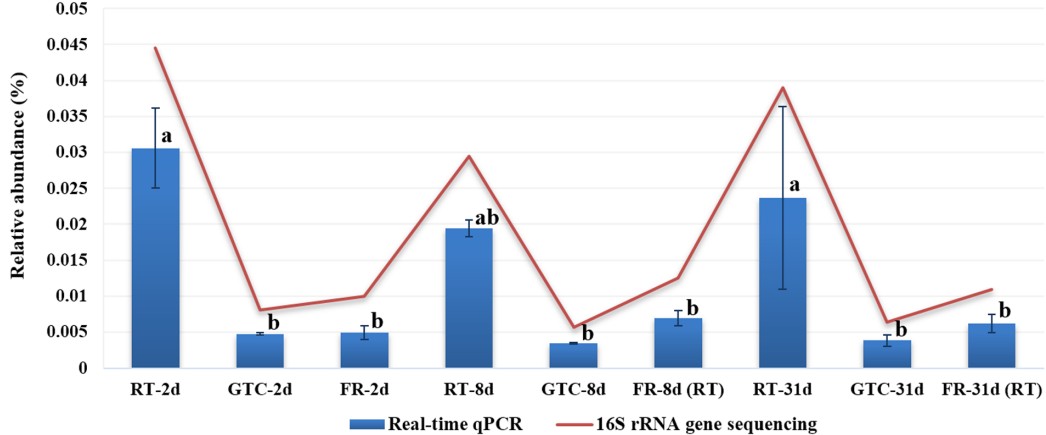

**Figure 4 Relative abundance of *Pseudomonas* spp. in all rhizosphere samples of peanuts, as determined by real-time qPCR and pyrosequencing.** FR-2d, rhizosphere samples stored at −70 °C for 2 days ($n = 3$). FR-8d (RT) and FR-31d (RT) are rhizosphere samples stored at −70 °C for 8 and 31 days, respectively, thawed at room temperature for 15 min before DNA isolation ($n = 3$). GTC-2d, -8d, and 31d are samples stored in 100 μL of GTC guanidine thiocyanate at room temperature for 24 h, and −20 °C for 1, 7, and 30 days, respectively ($n = 3$). RT-2d, -8d, and 31d are samples stored at room temperature for 2, 8, and 31 days, respectively ($n = 3$). The error bars represent standard deviation. [a,b,c]Different lower-case letters indicate significant differences between groups analyzed by one-way ANOVA (Tukey's HSD) analysis ($p < 0.05$).

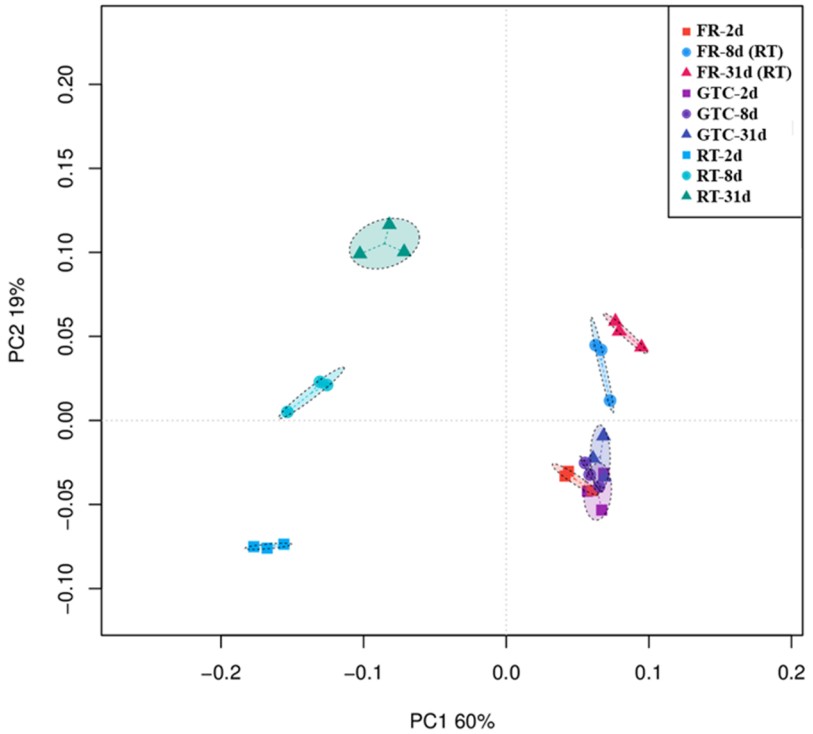

**Figure 5 Weighted UniFrac principal coordinate analysis (PCoA) of the rhizosphere bacterial community structure following the different treatments.** FR-2d, rhizosphere samples stored at −70 °C for 2 days (n = 3). FR-8d (RT) and FR-31d (RT) are rhizosphere samples stored at −70 °C for 8 and 31 days, respectively, thawed at room temperature for 15 min before DNA isolation (n = 3). GTC-2d, -8d, and 31d are samples stored in 100 µL of GTC guanidine thiocyanate at room temperature for 24 h, and −20 °C for 1, 7, and 30 days, respectively (n = 3). RT-2d, -8d, and 31d are samples stored at room temperature for 2, 8, and 31 days, respectively (n = 3). Ellipses, 95% confidence intervals.

## DISCUSSION

Many studies have shown that various plants (*Lundberg et al., 2012*; *Bulgarelli et al., 2013*; *Turner et al., 2013*) influence distinct rhizosphere bacterial communities, particularly beneficial bacteria, such as those that can promote biomass synthesis (*Rodriguez et al., 2008*), or increase resistance to bacterial (*Rudrappa et al., 2008*; *Lakshmanan et al., 2012*) and fungal plant pathogens (*Mavrodi et al., 2012a*, *2012b*). These studies have demonstrated that rhizosphere microorganisms may play important roles in future efforts to develop sustainable agriculture. To this end, investigation of the interactions between rhizosphere microbiome and plants is crucial.

In the present study, we developed a simple collection method using GTC solution to prevent changes in the bacterial community composition in plant rhizosphere samples at room temperature; this approach was shown to be an alternative method for the freezing of samples when sampling in the field. Pyrosequencing of 16S rRNA amplicons was used to evaluate the feasibility of the method, compare its effectiveness with that of freezing, and determine the reproducibility between different time intervals. The structure and OTUs of all GTC samples were shown to cluster together in the

PCoA analysis (Fig. 5), indicating that, in contrast to previously reported results (*Flores et al., 2015*), the removal of the GTC buffer before DNA isolation using a PowerSoil DNA Isolation Kit did not affect the bacterial composition of the sample and that no differences in rhizosphere bacterial community were found between samples treated with GTC and samples subjected to freezing.

To further analyze the effects of storage condition on the overall taxonomic compositions of rhizosphere samples and reproducibility of GTC preservation, Pearson's correlation coefficients of the relative abundance of all OTUs in all treatment groups were analyzed (Fig. S1). Remarkably, the relative OTU abundance in the GTC-31d sample showed high correlations ($r \geq 0.962$, $p < 0.01$) with those in the GTC-2d and GTC-8d groups. However, the OTU abundance at different time points in RT groups was shown to be less correlated ($r \leq 0.825$, $p < 0.01$). All abundances obtained for the GTC groups were shown to correlate more ($r \geq 0.909$, $p < 0.01$) with those determined in the FR-2d groups, than those of the RT groups ($r \leq 0.479$, $p < 0.01$). However, the correlation between the FR samples incubated for 15 min at room temperature before DNA extraction and the control samples ranged from 0.854 to 0.975 ($p < 0.01$).

This further illustrated that storing of the samples in GTC may represent an alternative to the frozen preservation of rhizosphere soil samples. Furthermore, the storage period in GTC did not affect the rhizosphere taxon profiles. This demonstrated that the reproducibility of the GTC-associated analyses was good. GTC preservation may help elucidate the bacterial community profiles of rhizosphere samples collected in the field.

According to our results, it is not recommended to store rhizosphere samples at room temperature regardless of the length of time. We found that the OTU number and alpha-diversity indices of the rhizosphere samples maintained at room temperature significantly decreased (Table 1, $p < 0.05$), most likely due to the degradation of bacterial DNA at room temperature. And the composition of these samples was also affected by changes in the abundance of individual taxa. Our findings are consistent with the majority of previous reports (*Rubin et al., 2013*; *Flores et al., 2015*). Thus, the thawed FR-8d and 31d samples even incubated at room temperature for 15 min prior to DNA isolation may lead to the alterations in the bacterial communities (Fig. 5; Fig. S1), suggesting that even for frozen samples, DNA should be extracted immediately after thawing.

## CONCLUSIONS

Taken together, our results established a method that allowed the samples to be stored at room temperature for at least 1 day prior to cryopreservation. GTC solution was found to effectively preserve rhizosphere soil samples at room temperature, eliminating the requirement of immediate freezing of rhizosphere soil samples during large-scale sampling in fields. This method did not affect the bacterial community structure and diversity of the rhizosphere soil. Accordingly, our novel approach may facilitate sample collection for plant rhizosphere microbiome analysis on a large scale and promote the investigation of plant–microbe interactions.

### Funding

This work was supported by the National Natural Science Foundation of China (Grant Nos. 31501711 and 31428020) and Science and Technology Innovation Project from the Chinese Academy of Agricultural Sciences (CAAS-XTCX2016012). The funders had no role in study design, data collection and analysis, decision to publish, or preparation of the manuscript.

### Grant Disclosures

The following grant information was disclosed by the authors:
National Natural Science Foundation of China: 31501711 and 31428020.
Science and Technology Innovation Project from the Chinese Academy of Agricultural Sciences: CAAS-XTCX2016012.

### Competing Interests

The authors declare that they have no competing interests.

### Author Contributions

- Xiaoxiao Sun performed the experiments, analyzed the data, prepared figures and/or tables, approved the final draft.
- Meiling Wang contributed reagents/materials/analysis tools.
- Lin Guo contributed reagents/materials/analysis tools.
- Changlong Shu contributed reagents/materials/analysis tools.
- Jie Zhang authored or reviewed drafts of the paper.
- Lili Geng conceived and designed the experiments, analyzed the data, prepared figures and/or tables, authored or reviewed drafts of the paper, approved the final draft.

### Data Availability

The V3 and V4 variable regions of 16S rRNA gene obtained from all samples are accessible via NCBI SRA database under accession number SRP134256.

### Supplemental Information

Supplemental information for this article can be found online at http://dx.doi.org/10.7717/peerj.6440#supplemental-information.

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
