# Peer review of "Guanidine thiocyanate solution facilitates sample collection for plant rhizosphere microbiome analysis"

_PeerJ, doi:10.7717/peerj.6440_

## Round 0.1 · original submission · Major Revisions

This manuscript was thoroughly reviewed by two experts in the field, as well as myself. Both reviews highlight the importance of finding new methods to preserve samples under field conditions, and thus see much value in this manuscript. However, both reviewers also point to numerous ways that the manuscript can be improved and rewritten. As suggested, I think an important step will be editing the manuscript for grammar, sentence structure, and clarity. Furthermore, while there is no need for new experiments, the manuscript will require significant changes in emphasis and interpretation and some reanalysis of the observed data to compare against standards for the field.

Reviewer 1 ·

Basic reporting

First, I think this method holds promise. Harvesting rhizosphere samples from individual plants can be a time consuming endeavor and one not particularly well-suited for field work. However, the introduction needs a better justification for why this method is needed. Lines 60-63 state that outdoor temperature is the foremost factor affecting the analysis of rhizosphere microbiota (note: the authors should provide a citation for this), but what does this mean and how does their method alleviate it? The authors need to clearly state what the potential problem is (e.g. is it time until freezing? temperature of field sampling?) and how their method may present a solution. In my opinion, the workflow to isolate the rhizosphere sample – harvest root, vortex, filter, centrifuge, remove supernatant - seems much more of a problem for field based rhizosphere studies than the inability to freeze immediately. In what circumstances would researchers have access to a vortex and a centrifuge capable of spinning 50 mL tubes at 3200 g but not a -20 C freezer? I would like to see a more thorough explanation of why field-based rhizosphere sampling is challenging in general and exactly how the method presented here may offer a solution to some, but not all, the existing challenges.

I think very clear questions in the final paragraph of the introduction will help improve the focus and clarity of the entire manuscript. I suggest these questions centre around the following objectives: 1) the optimization of the GTC protocol (i.e. the number of rinses with PBS prior to DNA extraction); and 2) the comparison of GTC samples with RT and FR samples. Questions related to objective 2 should explicitly state what exactly will be compared (i.e. the diversity within and between samples and the abundance of individual bacterial taxa).

In my opinion all text related to the peanut microbiome specifically (lines 74-76; 214-225; and 264-272), should be removed. As it is, this component of the study comes across as superficial and not particularly interesting. Lines 214-225 is simply a list of the OTUs found in common bacterial phyla in the peanut rhizosphere. This neither advances our understanding of the peanut rhizosphere specifically, nor the plant rhizosphere more generally, but instead distracts from the most important results.

Additionally, the English language should be improved to ensure that an international audience can clearly understand the text. For example, lines 231-233 could be improved as: “These studies demonstrate that rhizosphere microorganisms might play an important role in future efforts to develop sustainable agriculture”.

Several of the included figures seem unnecessary. Figures 3 and 4 exhibit redundancy, the authors should choose one figure to present the findings related to differential abundance. Figures 6 and 7 are redundant. The PCoA is much more intuitive and informative than the plot of relative abundance correlations. Finally, figure 8 is not terribly relevant and is redundant with figure 3.

Line 44: add “The” to the beginning of this sentence.

Line 47: Lugtenberg and Kamilova 2009 not in references.

Lines 66-67: Surely the ability of GTC to denature DNAse and RNAse is relevant here.

Line 69: What does “robust stability” mean? The authors need to be more precise with what measures of “stability” or “reliability” (line 163) mean. Does stable = no change in alpha diversity, no compositional changes, and/or no changes in the abundance of individual taxa? How long were fecal samples left at room temperature for in these cited studies?

Line 86: I think the authors want Bulgarelli et al. 2012

Line 164-166: This seems redundant with lines 93-95

Line 236: citation?

Lines 251-256: This is a very long sentence. Please split into at least two.

Experimental design

In general, the Materials and Methods need greater detail and some important points are completely missing. For example, why are the methods leading to the results presented in lines 151-161 not described or even mentioned in this section? Additionally, the results could be bolstered by more rigorous statistical methods:

1) Why not analyze the effect of sampling on alpha diversity indices (Table 1)? A simple ANOVA could be used here.

2) PERMANOVA can be used to assess whether compositional differences among sampling treatments are significant. See the Adonis function from the R package vegan for an example.

3) I’m not convinced that analyzing differences in the relative abundance of individual bacterial taxa across sampling treatments is the appropriate analysis (line 131 and 134). This is due to properties inherent to NGS datasets including compositionality and high sparseness (see Fernandes et al. 2014 Microbiome). There are now a number of sophisticated methods to analyze the abundance of bacterial taxa across experimental treatments, which take into account the compositional nature of microbiome datasets. For example the R packages DeSeq2 and ALDEx2 both do this.

Line 82: What is Huayu22? A peanut cultivar?

Line 87: How much root? Was it a standardized amount?

Line 91: What is 100-mesh? 100 microns?

Line 93: 900 mg of rhizosphere sample is a lot of material even from 9 plants. How did you obtain this much material from root samples that could only fit into 50 mL tubes? Additionally, how did you ensure that the 9 subsamples were homogenous?

Line 96: Why were the GTC samples not frozen at -70 C? The comparison between FR and GTC is made more difficult to interpret because of this difference.

Line 100: How were the frozen GTC samples treated? Surely some thawing took place with these samples as well? In general, I think most researchers would agree that some thaw is unavoidable when extracting DNA from frozen samples. In fact, when extracting DNA from many samples simultaneously, as is often the case, a 15 min thaw is very optimistic and my guess is that many samples thaw for greater than 30 mins. My point is the 15 min thaw, if only introduced for the FR samples, is a strange and potentially confounding treatment to add.

Line 108: The authors state they used a MiSeq here but on line 118 state they use a HiSeq.

Line 110: The authors need to cite the source where they obtained their barcodes and primer constructs or else provide them in the supplement.

Line 119: What is a tag (also in lines 121 and 122)? The authors until this point have used the term read or sequence, please keep the same terminology throughout.

Lines 119-120: The authors need a much better description of their sequence quality filtering. I have no idea what would be removed from this statement. Did the authors use Q scores to filter? What does “N in over 95% of positions and low-quality” refer to? Also please cite the authors of Trimmomatic.

Line 121: What is the Gold database?

Line 127: Greengenes database. Wang et al. 2007 is the citation for the RDP classifier, please cite DeSantis et al. 2006 for Greengenes.

Line 128: Already gave the version of USEARCH.

Line 129: Were raw read counts, rarefied read counts or relative abundance used to generate UniFrac distances? Please clarify and justify.

Line 130: The authors need to include package numbers, individual functions, and the R version used. Also, scripts used for USEARCH, Trimmomatic, and R would be very helpful.

Line 133: What does top 15 genera mean? Does this mean the 15 most abundant genera across GTC, RT, and FR-2 samples?

Validity of the findings

Many of the results were difficult to interpret due to a general lack of statistical rigor, see points 1, 2 and 3 in the above section of this review, Experimental Design. For example, lines 179-181 report that GTC samples had higher diversity than FR samples but how was this analyzed?

Lines 182-190: see point 3 above in Experimental Design.

Line 198: Switch rho to r if Pearson correlations were calculated.

Lines 205-212: see point 2 above in Experimental Design.

Line 229: How do rhizosphere bacteria induce plant photosynthesis?

Line 247: No statistical tests were performed on diversity indices, significance cannot be determined.

Lines 248-250: Can the authors speculate why storage at room temperature does not always cause perturbations to microbial community composition? Does it depend on sample type?

Lines 258-260: What is intraobserver reproducibility? I’m struggling to follow the logic in this statement.

Lines 274-279: The conclusions will be strengthened if the authors can return to clear questions asked in the introduction and state how their method will alleviate the challenges of rhizosphere sampling in the field. The authors state that this method will be helpful when many samples are collected but don’t explain how. Processing many rhizosphere samples in the field will not be made easier by skipping the fastest step (placing tubes in freezer), so really this method is most useful when researchers do not have access to a freezer in the field.

Reviewer 2 ·

Basic reporting

Sun et al. report on whether the treatment of peanut rhizosphere samples with guanidine thiocyanate (GTC) is a good way to preserve microbiome DNA in the field for later extraction and analysis. This is a useful question, and I expect many people working on plant microbiome studies will be interested in the answer. For the most part, Sun et al.’s report is professionally structured and written with appropriate figures and tables, and appropriate citations have been included. The raw data from this study have been made publically available.

My main concern about the basic reporting is that the manuscript was generally difficult to follow, due to a combination of minor grammatical errors throughout and some confusing structuring. I feel that the manuscript would benefit greatly from extensive copy editing to improve the grammar and structure. I have made a few suggestions along these lines in comments on the attached pdf.

In particular, I think the subject of the first paragraph of the Results (lines 150-161) should either be treated in the Methods, or be omitted/removed to supplemental information, and that background and citations given in the Discussion (lines 227-245) should appear in the Introduction instead. Also, samples FR-8d and FR-31d should be included in more of the text and figures.

Experimental design

The study is original primary research within the aims and scope of PeerJ. The research question is useful and timely, and the technical aspects of the study appear to be sound.

However, the experimental design does not fully address the research question as I understand it. The manuscript seems to be written as a test of GTC supplementation for preservation of intact microbial communities compared with sample storage at room temperature. The standard to which both GTC and room temperature storage are compared is cryopreservation (freezing at -70). GTC-treated samples were stored at room temperature for 24 hours and then frozen at -20 for an additional 1, 7, and 30 days before DNA extraction, but untreated samples were extracted after 2, 8, and 31 days at room temperature. Since no untreated samples were at room temperature for only 24 hours (and no treated samples were at room temperature for 2, 8 or 31 days), we can’t directly compare GTC addition and storage at room temperature. To put it another way, we can’t disentangle the contributions of freezing at -20 versus addition of GTC to the preservation of community profiles. I think a better test of the question as I understood it would be to compare samples treated with GTC and frozen after 24 hours with samples not treated with GTC then frozen after 24 hours, rather than left at room temperature for 2, 8, and 31 days.

Sun et al. demonstrate convincingly that GTC addition followed by 24 hours at room temperature prior to freezing at -20 does not significantly change the microbial community profile compared with immediate freezing at -70, which is a useful result. They also show that leaving samples at room temperature for 2 days and longer results in changes in the microbial community profile, which is also a good finding. However, I cannot tell from their study whether GTC addition preserves microbiome DNA any better than leaving samples at room temperature for 24 hours without any additive prior to freezing at -20. This manuscript should be rewritten to avoid giving the impression that GTC addition is directly compared with room temperature storage, because in my opinion the experimental design does not allow a meaningful comparison to be made.

Validity of the findings

The data from this study are publically available, and the findings appear to be valid in that the data are appropriately analyzed and visualized.

My main concern here is about comparing GTC treatment with room temperature storage for preservation of microbiome DNA, since I don’t think this is a valid comparison in this case (see comments above). Conclusions along these lines should be carefully rewritten to be consistent with what the results show.

Additional comments

I agree with the authors that immediate cryopreservation of samples in the field is usually not practical, and a good alternative for plant microbiome sample preservation is needed. One alternative practice is to store samples on ice while collecting in the field (eg. in a portable insulated container), and I wonder if the authors can comment on how they think this might affect microbial community profiles.

Annotated reviews are not available for download in order to protect the identity of reviewers who chose to remain anonymous.

---

## Round 0.2 · Minor Revisions

Thank you for resubmission of this manuscript to PeerJ. One previous reviewer and I have evaluated this resubmission, and we both feel it is substantially improved. However, there remain a couple of small changes that need to be addressed prior to acceptance. My suggestions are as follows, but please also note those of reviewer 1.

L38: “to promote” better as “that promote”

L61-75: would be good to include a sentence here about how these different conditions might bias results

L78: better as “Guanidine isothiocyanate denatures RNAse and DNAse”

L188: “The copy numbers of the 16S rRNA gene in these samples were” better as “Copy number of the 16S rRNA gene transcript were”

L189: better as “did not affect copy number of the 16S rRNA gene transcript”

L191: better as “rRNA gene transcript”

L195 and throughout: would be good to include “, “ with larger numbers.. So 56214 better as 56,214.

Reviewer 1 ·

Basic reporting

I commend Sun et al. for producing a nicely revised manuscript. The introduction lays out the objectives of the study more clearly now. However, there are still some critical methods missing.

Line 40: Citation needed here

Line 76: Please state what ‘promotes stability’ actually means here(e.g. no change in diversity or composition?).

Line 107: Please state that 100 mesh = 100 openings per square inch.

Line 172-174: Please state where this universal primer pair comes from.

Line 215: please change to p < 0.001, or something similar.

Line 219: This was supposed to be change to ‘illumina sequencing’.

Line 227-228: In order for the ANOSIM results to be interpretable, the reader needs some description of the analysis. Please include this in the methods. Readers will not know what R means in this context.

Figure 6: This is not interpretable. What is the y-axis? I recommend removing this figure. Fig. 5 is a nice representation of the compositional differences among sample types and the ANOSIM provides statistical support. Fig. 6 does not help with the comprehension of this result.

Line 267: It seems likely that storing rhizosphere samples at room temperature affects diversity and composition not only through DNA degradation but also through changes in the abundance of individual taxa. Tube at room temperature is a very different niche than the rhizosphere and likely selects for particular bacterial clades.

Line 277: Please remove subsequent analysis from this sentence and simply state that the method did not affect bacterial diversity or structure.

Table 1: The results from the significance tests don’t make a lot of sense for High-Quality Sequences. FR-8d RT and RT-2d are nearly identical in mean and variation but are significantly different? While FR-2d appears to be quite different yet is not significantly different. Please double check these results. Are values mean plus/minus standard error? Please state in caption. Also please be consistent with the rounding of numbers i.e. some values have a decimal place while others do not. Finally, this is the first I see any mention of stats (ANOVA – Dunnett’s T3). Please describe these analyses in the appropriate section, including why Dunnett’s test was performed.

Experimental design

Line 183-193: The statistical analyses used to obtain the results presented here are not described anywhere. Please provide a description of the analyses used in the ‘GTC Buffer and DNA extraction’ section.

Line 202: The statistical analyses used to obtain the results presented here are not described anywhere. Please provide a description of the analyses used in the ‘downstream data analysis’ section.

Validity of the findings

no comment

---

## Round 0.3 · Minor Revisions

Thank you for your resubmission to PeerJ. I've gone through the manuscript and there are a a couple of things I feel need to be corrected before final acceptance.

1) In every figure legend where appropriate (Figs. 2,3,4), please include in the legend what the error bars stand for. +/- 1 standard deviation?

2) I don't think this will change much, but for analysis of the data in figures 2 and 4 you use Fisher's LSD test to group as a post-hoc ANOVA test. Fisher's LSD really isn't reliable when there are more than 3 groups, however, so you should reanalyze these groups using something like Tukey's HSD. I don't think it will make much of a difference, but if it does, please just also go through the text and make sure your interpretation in the text matches the data when analyzed with Tukey's HSD.

---

## Round 0.4 · accepted · Accept

Thank you very much for the resubmission, and for making the required changes. Congratulations on your new paper!

#